# Influence of Psychosocial and Sociodemographic Variables on Sickness Leave and Disability in Patients with Work-Related Neck and Low Back Pain

**DOI:** 10.3390/ijerph17165966

**Published:** 2020-08-17

**Authors:** Israel Macías-Toronjo, José Luis Sánchez-Ramos, María Jesús Rojas-Ocaña, E. Begoña García-Navarro

**Affiliations:** 1Department of Rehabilitation, Huelva Fremap Hospital, 21003 Huelva, Spain; israel.macias123@alu.uhu.es; 2Department of Nursing and Health Sciences, University of Huelva, 21007 Huelva, Spain; jsanchez@uhu.es (J.L.S.-R.); bego.garcia@denf.uhu.es (E.B.G.-N.)

**Keywords:** musculoskeletal disorders, low back pain, neck pain, sickness leave, psychosocial pain, catastrophizing, kinesiophobia

## Abstract

The purpose of this study was to describe the association between psychosocial factors in patients with work-related neck or low back pain (*n* = 129), in order to study sickness leave, its duration, the disability reported, and to analyze the relationship of these factors with different sociodemographic variables. This was a descriptive cross-sectional study. Data on kinesiophobia, catastrophizing, disability, and pain were gathered. Sociodemographic variables analyzed included sex, age, occupational, and educational level. Other data such as location of pain, sick leave status and duration of sickness absence were also collected. Educational level (*p* = 0.001), occupational level (*p* < 0.001), and kinesiophobia (*p* < 0.001) were found to be associated with sickness leave; kinesiophobia (b = 1.47, *p* = 0.002, r = 0.35) and catastrophizing (b = 0.72, *p* = 0.012, r = 0.28) were associated with the duration of sickness leave. Educational level (*p* = 0.021), kinesiophobia (b = 1.69, *p* < 0.000, r = 0.505), catastrophizing (b = 0.76, *p* < 0.000, r = 0.372), and intensity of pain (b = 4.36, *p* < 0.000, r = 0.334) were associated with the degree of disability. In the context of occupational insurance providers, educational and occupational factors, as well as kinesiophobia and catastrophizing, may have an influence on sickness leave, its duration and the degree of disability reported.

## 1. Introduction

The incidence of neck and low back pain has risen on a global level, not only in industrialized, but also in developing countries, with ensuing social, healthcare, and employment costs [1,2]. In Spain, the prevalence of both neck pain (NP) and low back pain (LBP) has increased in recent years [3]. Worldwide, figures have gone up in the last few years, with the prevalence of acute LBP throughout an individual’s lifetime standing at 84%, and that of chronic LBP at 23% [4]. Globally, one-year prevalence ranges between 17% and 75% for NP and from 22% to 65% for LBP [5], occupying the first and second places, respectively, of the most severely disabling conditions [5,6]. In Spain, LBP and NP constitute the most common causes of short-term sickness absence and may result from an occupational accident sustained either at work or en route to or from work [7].

Some authors report that 37% of cases of LBP worldwide could be attributed to factors related to the workplace environment [8,9]. Occupations in which workers are exposed to vibrations or are required to adopt a forced posture or to lift weights seem to be more likely to result in the appearance of episodes of NP and LBP [10,11]. As regards to social and occupational factors, the evidence shows that low levels of social coverage, job dissatisfaction, and physically demanding jobs could be associated with the development of LBP [8].

On a psychological and a social level, kinesiophobia and fear-avoidance behaviors appear to be among the factors which play the greatest role in the evolution of musculoskeletal pain and its transition toward chronicity [12,13]. Kinesiophobia and hypervigilance to pain behaviors would seem to be based on catastrophizing thoughts, which activate self-limiting attitudes, which in turn, amplify disability and pain [12]. Consequently, catastrophizing thoughts are related to fear of movement, which is connected to poorer results in terms of the prognosis of medical outcomes [14,15]. 

Clinical guidelines on musculoskeletal disorders and, more specifically, neck and low back pain, insist on the need to identify psychosocial risk factors as part of a multidisciplinary approach to the management of these patients [2,16,17]. The influence of pain-related fear of movement and catastrophizing on sickness leave in patients who have sustained a musculoskeletal injury has been scarcely studied from the point of view of an occupational insurance provider. There are published studies that analyze back pain from the point of view of primary care within public health care [18,19,20]. However, the literature is lacking in reports that take the perspective of an occupational insurance provider, which is surprising in view of the fact that, in many European countries, occupational conditions are managed specifically by occupational insurance providers as collaborators of social security systems. In Spain, occupational health insurance providers are private companies that collaborate with the Public Social Security System and are responsible for managing economic compensations, prevention, and health assistance in the event of occupational disease. Thus, patients on sick leave due to an accident at work receive from an occupational health insurance provider economic benefits which cover at least 75% of their salary [21]. In this sense, occupational health care is considered specific and agile in terms of resources such as specialist medicine, complementary tests, or rehabilitation treatment in comparison with public health services. Considering that kinesiophobia and catastrophizing appear to have an impact on patients’ recovery process and that they could in turn influence the outcome of therapeutic approaches, this article seeks to determine the relationship between these strictly non-clinical conditions and the progression of work-related neck and low back pain, emphasizing the importance of taking them into consideration as part of an overall treatment strategy.

Therefore, the purpose of this study was to investigate the association between kinesiophobia and pain-related catastrophizing as psychosocial factors in patients with work-related neck and low back pain; to analyze the influence of these factors on sickness leave, its duration, and the degree of disability reported; and to look into the relationship between kinesiophobia and catastrophizing across a series of sociodemographic variables.

## 2. Materials and Methods

The methodological strategy was planned with a descriptive observational design, placing the study in the Clinical Health Service of an occupational insurance provider.

### 2.1. Subjects

The subjects of the study were individuals (*n* = 129) with LBP or NP who presented to an occupational health clinic with a diagnosis of non-specific work-related low back and neck pain due to a work accident between 1 June 2018 and 31 December 2019. Non-specific pain is considered to be pain that is not caused by fractures, direct trauma, or systemic disease and where there is no proven root compression amenable to surgical treatment. A work-related accident is any bodily injury that the employed worker suffers on the occasion or as a result of work [21].

On the same day of the accident, injured patients were attended by the occupational insurance medical service and, after diagnosis and typification as an occupational accident, it was noted whether patients had a sickness leave status or whether they could reconcile their neck and low back pain with their professional activity. On the first or second day after the accident, the patients were informed about the purpose of the study and signed the relevant informed consent form and completed questionnaires. The tracking of the duration of the sickness leave was followed up and measured on the days off from the day of the work accident until the return to work.

This study followed the ethical principles for medical research in human beings according to the Declaration of Helsinki and the protection of data and guarantees of digital rights according to organic law 3/2018 of 5 December 2018. The study was authorized by the Fremap Mutual Ethics Committee (Code number FREMAP-2200631-Z).

#### Selection Criteria

All patients who met the following inclusion criteria were consecutively included:

*Inclusion Criteria*
Patients with work-related non-specific low back pain and/or neck pain presenting to an occupational health clinic.Age between 18–65 years.Being able to understand Spanish; having signed an informed consent form.

*Exclusion Criteria*
Infection-related neck and low back pain, cancer, fracture, visceral disease, spondylarthritis, extruded disc herniation, cauda equina syndrome.Previous treatment for neck and low back pain.Previous surgery, commuting accident, work-unrelated conditions.Cognitive impairment.

### 2.2. Study Variables

#### 2.2.1. Clinical Variables

##### Tampa Scale for Kinesiophobia

Fear of movement or re-injury is considered to be a predictor of pain perpetuation and is one of the main constructs in the Cognitive-Behavioral Fear-Avoidance Model. The Tampa Scale for Kinesiophobia [22] is one of the main tools used by the model to measure a subject’s fear of movement or re-injury during motion. Although the Tampa Scale for Kinesiophobia was originally used to assess fear of movement derived from LBP, it has shown its validity from a psychometric perspective in patients with NP [23]. The 11-item Tampa Scale for Kinesiophobia used in this study showed itself to be as valid and as reliable as the original longer 17-item scale [24]. The scale’s Spanish version has been validated with good results for patients with both chronic and acute pain [22]. The scale consists of 11 items and each item is scored based on a 4-point Likert scale, ranging from “strongly agree” (4 points) to “strongly disagree” (1 point). The overall score ranges from 11 to 44, with the highest scores indicating a strong fear of movement or re-injury [24].

##### Pain Catastrophizing Scale

Although pain-related catastrophizing is considered a significant prognostic factor in chronic pain, it is also often used as a prognostic factor in individuals with acute pain [20]. The Pain Catastrophizing Scale (PCS) is a 13-item scale that measures the extent to which subjects develop certain feelings and thoughts related to their nociceptive experience [25]. It consists of three scales: rumination, magnification, and helplessness. This instrument, which has a version validated for Spanish [26,27], has been shown to have adequate internal consistency (Cronbach’s alpha = 0.818 for athletes and 0.79 for patients with fibromyalgia) and an intra-class correlation of 0.84 for patients with fibromyalgia. The PCS has been validated for neck and low back pain in other languages than Spanish [28,29,30]. This study considered the overall score, not the scores on separate subscales.

##### Roland Morris Disability Questionnaire

This instrument was used to measure lower back disability. It is one of the world’s best validated and most commonly used scales to measure LBP-related disability. It consists of 24 items related specifically to daily physical activities likely to be affected by LBP. A score between 0 (no disability) and 24 (maximum disability) is obtained by adding up the number of checked items. A Spanish version has been validated and has become a reliable instrument to evaluate disability in patients with LBP [31], with an intra-class correlation of 0.87, good concurrent and construct validity, and high internal consistency (Cronbach’s alpha = 0.837 for the 1st day and 0.914 for the 15th day). LBP-related disability is expressed in absolute terms.

##### Neck Disability Index

Neck disability level was measured by means of the Neck Disability Index, a 10-item questionnaire that asks patients about the effects of their NP on the performance of a series of everyday activities. For each item, respondents are given 6 possible answers, which they are asked to score from 0 (no activity limitations) to 5 (major activity limitations). The instrument, which has been translated into Spanish, is reliable and has enough internal consistency to measure disability in patients with NP [32]. NP-related disability is expressed in absolute values.

Disability was measured with both the Roland Morris Disability Questionnaire for LBP and the Neck Disability Index for NP on a global standardized scale made for the purpose.

##### Numeric Pain Rating Scale

The Numeric Pain Rating Scale was used to measure intensity of pain at the time of the intervention [33]. On an 11-item scale, 0 indicates an absence of pain while 10 represents the worst imaginable pain.

#### 2.2.2. Sociodemographic Variables

Sociodemographic data such as sex, age, occupational, and educational levels were gathered as well as data on pain location, sick leave status, and duration of sickness absence (in days).

The 2011 Spanish National Occupations’ Classification put together by the Spanish Office for National Statistics, was used for the “type of work” variable. This classification rates occupations according to the level of competences required. It includes 4 levels [34]:Competence level 1. Occupations under competence level 1 usually entail the performance of simple and routine physical or manual tasks. Use of manual tools may be required.Competence level 2. Occupations under competence level 2 usually require performing tasks such as machine and electronic equipment operation, driving of vehicles, maintenance and repair of electrical and mechanical equipment, and handling, organizing, and sorting information.Competence level 3. Occupations under competence level 3 tend to encompass technical tasks and complex routines that require different kinds of technical and practical knowledge specific to a certain subject.Competence level 4. Occupations under competence level 4 require the performance of tasks involving decision-making and complex problem-solving based on a profound theoretical and practical understanding of a certain subject matter.

For the educational level variable, subjects were assigned to different groups depending on whether they had completed primary education, secondary education, pre-university education, or university education.

### 2.3. Statistical Analysis

Firstly, a descriptive analysis was made of the above-mentioned variables. Means and standard deviations (SD) were included for continuous variables and absolute and relative frequencies for categorical variables. To analyze the relationship between variables, a description was made by which a bivariate analysis was conducted of variables associated with sickness leave, its duration, and the degree of disability reported. The relationship between sociodemographic variables (sex, education, occupation, and location) and sickness leave was analyzed by means of chi-squared testing once the application conditions were verified. The professional competence levels were reclassified, grouping the only case under level 4 together with those of level 3. After making sure of the absence of deviations by means of a detrended Q–Q plot and variance homogeneity, the means of the clinical variables and age were compared using Student’s t testing of the hypotheses of no differences between patients who presented sickness leave status and those who did not.

The association of clinical variables with the duration of sick leave and the degree of disability was evaluated using a linear regression model controlling for the variables of occupation and educational level. The association of the variables kinesiophobia, catastrophizing, and pain intensity with the need for sick leave was explored using multiple logistic regression, controlling for sociodemographic variables. The association of the variables kinesiophobia, catastrophizing, and pain intensity with the duration of sickness leave and the level of disability declared was explored using multiple linear regression, controlling for the sociodemographic variables.

## 3. Results

The present study recruited 129 subjects with work-related NP and LBP for whom 71 were male (55%) and 58 were female (45%). Most cases corresponded to low back pain (68.2%), with neck pain corresponding to a minority of cases (31.8%). Subjects with primary education were in the majority (48%). Other educational levels accounted for decreasing proportions of the sample, with subjects with a university education constituting the smallest group (10.1% of the cohort). A similarly decreasing pattern was observed in the realm of occupational competence, where 106 subjects (82%) were classified as level 1 and only 1 (0.8%) as level 4. To increase comparative efficiency, the only subject under level 4 was grouped with those under level 3. In the majority of cases, work-related neck and low back pain resulted in sickness leave, with 62.8% of subjects being absent from work at least one day as compared with 37.2% who reported to work in spite of their condition. Mean age was 40.1 (SD = 9.43) years. Mean duration of sickness absence was 23.74 (30.27) days. In most cases, the degree of disability declared at the first medical contact was 45.38%, with the number of low back pain reports slightly exceeding those of neck pain (46.64% vs 42.67%). The mean score on the Tampa Scale for Kinesiophobia was 30.53 (71.1). The mean score on the Pain Catastrophizing Scale was 27.66 (11.69) for all subjects. The mean score on the Numeric Pain Scale was 7.02 (1.83) for all subjects.

The Tampa Scale for Kinesiophobia–Pain Catastrophizing Scale (r = 0.48), Tampa Scale for Kinesiophobia–Numeric Pain Scale (r = 0.30), and the Pain Catastrophizing Scale–Numeric Pain Scale (r = 0.39) correlations, although significant, are relatively weak. In any case, the final analysis was carried out adjusting for the three variables.

### 3.1. Sickness Leave

The educational and occupational level sociodemographic variables appeared to be related with sickness leave variable in Table 1. As compared with those with a university education, subjects with secondary education had a 4.44 times higher chance of going on sickness leave (CI 95%: 1.04–19.02). The chance in subjects with primary education was 11.67 higher (CI 95%: 2.82–48.28). Subjects under occupational level 1 were 9.68 times more likely to go on sickness leave than those at occupational level 3 (CI 95%: 2.55–36.69), and in subjects under occupational level 2 sickness leave was 2.40 times more likely than in those at level 3 (CI 95%: 0.36–16.21). According to the data gathered in this study, sex and pain level did not seem to be associated with sickness leave. Nor did the age variable seem to be associated with sickness absence; subjects on sickness leave had a mean age of 40.33 (9.36) years, while those who were not on leave had a mean age of 39.77 (9.61) years.

As regards the clinical variables, Table 2 describes the relationship between suffering from kinesiophobia and sickness leave. Subjects on sickness leave exhibit a higher mean score on the kinesiophobia scale (32.27 (7.18)) than those not on sickness leave (27.58 (6.00)) (*p* < 0.001). Table 3 (multiple logistic regression) shows that the Tampa Scale for Kinesiophobia appeared to be related with sickness absence, controlling for educational level and occupational level as confounding variables.

### 3.2. Duration of Sickness Leave

We found no sociodemographic variables related to the duration of sickness absence. However, using a linear regression model, kinesiophobia (b = 1.47, *p* = 0.002, r = 0.35) and catastrophizing (b = 0.72, *p* = 0.012, r = 0.28) did appear to be associated with the duration of leave, the association with kinesiophobia being stronger in a model where the different clinical variables were individually controlled for educational level and occupational level. Contrary to what may be expected, intensity of pain (b = 2.78, *p* = 0.208, r = 0.14) did not appear to be associated with the duration of sickness leave as described in Table 4.

### 3.3. Disability

Educational level appeared to be related to the degree of disability (*p* = 0.021)*,* the association between occupational level and disability does not achieve statistical significance (*p* = 0.081) (linear regression). Kinesiophobia (b = 1.69, *p* < 0.000, r = 0.505), catastrophizing (b = 0.76, *p* < 0.000, r = 0.372), and intensity of pain (b = 4.36, *p* < 0.000, r = 0.334) were seen to be clearly associated with the degree of disability, controlling for the educational level and the occupational level as confounding variables. Accordingly, in a multivariate clinical regression model, kinesiophobia and pain retained their association with disability both in subjects in active employment and in those on sickness leave as described in Table 5. Catastrophizing did not exhibit an association with the degree of disability reported in any of the models studied. Consequently, reported intensity of pain and kinesiophobia had a clear association with the level of disability reported at the first medical contact.

## 4. Discussion

The purpose of this study was to understand the influence of psychosocial variables and of intensity of pain on sickness leave, duration of sickness absence, and reported degree of disability, as well as their relationships with different sociodemographic variables in the context of an occupational health clinic treating work-related diseases within the Spanish Healthcare System.

### 4.1. Sociodemographic Variables

Both occupational and educational factors have previously been shown to be associated with neck and low back pain [35], long-term persistence of pain [36], and less successful medical interventions [37,38]. In this respect, although this study did find an association between these sociodemographic factors and sickness leave status, no such association was observed between the duration of sickness absence and the educational and occupational level variables, which contrasts with the findings of the studies mentioned above. Factors such as the more thorough patient follow-up that is typical of an occupational health clinic (as compared with a public hospital) and the greater availability of complementary tests and/or alternative treatments such as physical therapy, or even surgical procedures, could be related to the shorter sickness absence observed in patients treated at an occupational health clinic. Accordingly, at a first consultation following an occupational accident where health care professionals basically determine the subject’s sickness leave status, educational and occupational variables would be associated with whether sickness leave status is granted, but not with its duration.

The relationship between reported degree of disability, occupational level, and socioeconomic factors have been studied in the literature [37,39,40], also finding a positive association between occupational and educational factors and disability. This study shows that the educational level holds a stronger relationship with disability than the occupational level, which contradicts previous reports on the subject. Accordingly, clinical variables were controlled for educational and occupational level.

### 4.2. Clinical Variables

Negative emotional states such as kinesiophobia and pain-related catastrophizing have been associated with sick leave due to neck and low back pain [41,42] and are considered a poor prognosis factor in patient recovery [43]. In this paper, the clinical variable of kinesiophobia appears to be strongly correlated with sickness absence and with the duration thereof, but not with the intensity of pain reported at the first medical contact. In the work of other authors, fear of movement has been associated with poorer results in terms of recovery from episodes of neck and low back pain, a finding that should be considered in the context of interventions related to occupational accidents [44] regardless of the intensity of pain [45]. The finding that there is an association between kinesiophobia and the duration of sickness absence is in line with previous reports on neck pain conditions [46]. Fear of movement and re-injury is a factor that would appear to slow down recovery in these patients.

Intensity of pain, on the other hand, has not been shown to be associated with whether sickness leave is granted or not, or with its duration. Consequently, the return-to-work rate would not appear to be related to the intensity of pain reported at the first medical contact, which is in line with previous reports on the subject [47]. Unlike other studies, the origin of follow-up for these patients started from the day of the workplace accident or the next, which would explain the high levels of pain reported and why the acute nature of the pain reported in the first care would not be related to the sickness leave status or its duration.

This study found no association between pain catastrophizing levels and sickness leave status. It did, however, find that in patients with neck and low back pain, pain catastrophizing was associated with the time it took subjects to return to work. Again, negative thoughts and hypervigilance with respect to the condition could delay these patients’ return to work. In this regard, this study echoes the findings of other authors who have demonstrated that expectations, pain catastrophizing levels, and fear of movement play a role in patient recovery and in the duration of sickness [43,47,48]. In our context of occupational insurance providers, it is the medical staff who make the initial diagnosis, the person in charge of determining the sickness leave status of the patients and, in this sense, these personal catastrophic thoughts would not intervene in a medical decision about the capabilities of a patient to do his job. Instead, it seems that these catastrophic thoughts would be related to the duration and chronification of symptoms in a sickness leave status. Lastly, the findings of this paper indicate that fear of movement, the degree of catastrophizing and the intensity of pain are related with disability, which is in line with other reports on acute patients [49]. Under the Fear Avoidance Model, fear of movement is associated with disability, and some authors point to pain catastrophizing as a key element in the relationship between disability and sickness absence [20], which contrasts with the findings of this study. Indeed, in this paper, kinesiophobia, fear of re-injury and intensity of pain have a much stronger association with disability. According to the literature, factors such as hypervigilance, expectations and pain catastrophizing are constructs that form prior to the development of kinesiophobia, which predetermine the degree of kinesiophobia experienced by an individual. However, the nature of this sequence cannot possibly be established by a cross-sectional study like the present one. Nonetheless, our model is comparable to others proposed in the past where the relationship between pain and disability would seem to be mediated by fear of movement [50]. In fact, this paper found a strong association between kinesiophobia, pain and degree of disability.

In any event, the expectations preceding a work-related accident, pain catastrophizing, fear of movement and the intensity of pain seem to be associated with the degree of disability reported at the first medical contact, with the relationship between fear of movement and pain and disability being the strongest.

In our work, we found a greater association of fear of movement on the variables studied compared to pain catastrophizing. In this sense, other authors have found similar results where fear of movement appears as the main factor in returning to work and disability above other psychosocial variables of coping strategies, including pain catastrophizing [51,52]. Thus, authors insist on the relationship of pain catastrophizing and the ability of subjects to generate positive thoughts about problem solving in the context of chronic pain [53]. In our setting, where the recruitment of patients is made from the beginning of the accident and pain is in the context of acute pain, fear of movement is presented as the main factor that would determine the sickness absence status, its duration and the disability reported. We did not find studies where, as in ours, the recruitment of the subjects has been done the same day or the day after the work accident. In this sense, the short space of time from the first symptoms and the measurement carried out could have an important role in the results obtained. As we have already mentioned, not only high levels of pain intensity could be related to the measurement time of the first symptoms, but this relationship could also occur in the rest of the variables studied. Likewise, the results obtained in such an early measurement, prior to any therapeutic intervention, would have to be taken into account to implement strategies that include these type of psychosocial variables that would be associated with the evolution of neck and low back pain processes.

Although this cross-sectional study followed up on the duration of sickness absence among the patients in the sample, it does not warrant the establishment of a cause–effect relationship with respect to the variables analyzed. In this sense, longitudinal studies would be recommended.

It is the hope of the authors of the present study that its findings may encourage the performance of further studies on the same subject matter with larger cohorts, and might including subjects from different employment backgrounds.

## 5. Conclusions

Taking into account that the workplace-related accident is specifically managed by the occupational health insurance providers, not only in the health but also in the economic aspect, and that the scientific literature does not report studies from this perspective, this work is original in its attempt to relate, in a work context, educational and occupational factors with sickness leave and the degree of disability reported, but not with the duration of neck and low back pain-related sickness leave.

The more severe forms of kinesiophobia derived from work accidents are associated with a greater likelihood of being on sickness leave due to neck and low back pain, with a greater likelihood of longer periods on leave, and with more severe disability. Pain catastrophizing does not appear to be associated with the disability itself but with a longer sickness absence from work and more severe disability. Finally, intensity of pain does not seem to be related to the likelihood of being on sickness leave or with its duration, but it is, however, related to a more severe level of disability being reported at the first medical assistance.

In this sense, these types of psychosocial factors, which are scarcely taken into account in the recovery from an occupational accident process, should be included in guidelines and approaches for this type of patient following an occupational accident, in order to facilitate patients’ recovery and speed up their return to work. It must be noted that this type of variable is not normally included in the healthcare protocols used to approach these kinds of conditions.

## Figures and Tables

**Table 1 ijerph-17-05966-t001:** Frequency of sickness leave in each sociodemographic group. † Chi-square test *.

Variables	Sickness Leave (n)	Sickness Leave (%)	† *p*	Comparison	Odds Ratio	Inferior Limit CI 95%	Superior Limit CI 95%
**Sex**							
Males Females	45 36	63.4 62.1	0.878	Men/women	1.06	0.52	2.17
**Location**							
Neck Lower back	21 60	51.2 68.2	0.063	Lower back/Neck	0.49	0.23	1.05
**Education**							
Primary Secondary Pre-university University	49 20 9 3	77.8 57.1 50.0 23.1	0.001	Primary/University Secondary/University Pre-University/University	11.67 4.44 3.33	2.82 1.04 0.68	48.28 19.02 16.30
**Occupation ****							
Level 1 Level 2 Level 3–4	75 3 3	70.8 37.5 20.0	<0.001	Level 1/Level 3–4 Level 2/Level 3–4	9.68 2.40	2.55 0.36	36.69 16.21

* Performing a sensitivity analysis in which the sickness leave status variable is required to last a minimum of 8 days, the association between sociodemographic and sickness leave status variable is maintained, as presented in Appendix A. † Chi-square test. ** Occupations: Level 1: Simple and routine physical. Level 2: Handling of machinery, electronic equipment, storing and sorting information. Level 3: High educational level, advanced communication skills, ability to understand complex written materials. Level 4: performance of tasks involving decision-making and complex problem-solving based on a profound theoretical and practical understanding of a certain subject matter.

**Table 2 ijerph-17-05966-t002:** Mean values for the different clinical variables as a function of the presence (yes) or absence (no) of sickness leave status. ‡ t-Student.

Sickness Leave	Yes			No			‡ *p*
	n	Mean	SD	n	Mean	SD	
Catastrophizing Scale	81	28.76	11.71	47	25.75	11.51	0.159
Kinesiophobia Scale	81	32.27	7.18	48	27.58	6.00	0.000
Intensity of Pain	81	7.23	1.55	48	6.69	2.19	0.101

**Table 3 ijerph-17-05966-t003:** Multiple logistic regression model: influence of kinesiophobia on sickness leave, controlled for educational and occupational level.

Sickness Leave	Odds Ratio	CI 95% Odds Ratio		*p*
		Lower	Upper	
Kinesiophobia Scale	1.09	1.03	1.16	0.006
Education Scale				0.328
Primary/university Secondary/university Pre-university/university	2.73 1.16 1.48	0.19 0.08 0.12	39.70 17.09 18.76	0.462 0.914 0.764
Occupation				0.352
Level 1/Level 3 Level 2/Level 3	4.60 2.13	0.39 0.13	54.46 33.97	0.226 0.594

**Table 4 ijerph-17-05966-t004:** Association of the duration of absence variable with the clinical variables analyzed in this study, controlling for educational and occupational level.

Duration of Sickness Leave	b	95% CI (b)		r	*p*
		Lower	Upper		
Catastrophizing Scale	0.76	0.16	1.29	0.28	0.012
Kinesiophobia Scale	1.47	0.58	2.36	0.35	0.002
Intensity of Pain	2.78	−1.59	7.15	0.14	0.208

**Table 5 ijerph-17-05966-t005:** Multivariate linear regression between reported degree of disability and kinesiophobia, intensity of pain and catastrophizing scales.

**(a) Initial Models, Adjusted by Educational and Occupational Level**					
	**b**	**95% CI (b)**		**r**	***p***
		**Lower**	**Upper**		
**Catastrophizing Scale**	0.76	0.41	1.11	0.37	0.000
**Kinesiophobia Scale**	1,69	1.18	2.21	0.51	0.000
**Intensity of Pain**	4.36	2.18	6.54	0.33	0.000
**(b) Final multivariate model, adjusted for educational and occupational level and for the remaining variables in the model**					
	**b**	**95% CI (b)**		**r**	***p***
		**Lower**	**Upper**		
**Catastrophizing Scale**	0.25	−0.12	0.62	0.12	0.183
**Kinesiophobia Scale**	1.34	0.77	1.91	0.40	0.000
**Intensity of Pain**	2.46	0.34	4.57	0.19	0.023
**(c) Final multivariate model, adjusted for educational and occupational level and for the remaining variables in the model for patients on sickness leave.**					
	**b**	**95% CI (b)**		**r**	***p***
		**Lower**	**Upper**		
**Catastrophizing Scale**	0.31	−0.16	0.77	0.15	0.192
**Kinesiophobia Scale**	1.08	0.37	1.79	0.33	0.004
**Intensity of Pain**	3.82	0.68	6.96	0.25	0.018

(**a**) individual regressions for each variable; (**b**) final model: multiple regression including all the clinical variables; (**c**) multiple regression including all the clinical variables in patients on sickness leave. All models were controlled for educational and occupational level.

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
