# Peer review of "Influence of Psychosocial and Sociodemographic Variables on Sickness Leave and Disability in Patients with Work-Related Neck and Low Back Pain"

_ijerph, 2020, doi:10.3390/ijerph17165966_

Round 1

Reviewer 1 Report

Well done article.

moderate English changes required.

The bibliografy should be expanded and the tables clarified.

Reviewer 2 Report

This study is a descriptive cross-sectional study and was to examine the association between psychosocial factors in patients with work-related spinal pain, to examine sickness leave its duration, disability reported, and to analyze the relationship of these factors with different sociodemographic variables. Data on kinesiophobia, catastrophizing, disability, and pain were gathered and sociodemographic variables analyzed included sex, age, occupational and educational level. The main contribution of this study was to suggest some factors should be taken into account when treating spinal pain following an occupational accident, in order to facilitate patients’ recovery and speed up their return to work. In general, the manuscript is well written and clearly described. My most concern is that the sample size (n=129) is relatively small compared to the similar previous studies, even though this is essentially a cross-sectional study. This smaller sample size would affect the statistical significance and power and should be highlighted in the study limitation. Some specific comments for the manuscript are also as follows.

L27, please define what's the spinal pain. In this study, only NP and LBP were introduced.

L9, abstract, the sample size should be mentioned in the Abstract. This information is very important.

L29, “henceforth” can be removed.

L59, patients on sick leave due to an accident at work receive from occupational health insurance provider an economic benefit that covers at least 75% of their salary. Please cite the reference or website.

L76, Subsection, ex. “2.1 Type of study” is not necessary to be indented. The rest are inferred analogy.

L77, how to convince that there was no prescreening or Hawthorne effect happened for this descriptive observational design.

L83, Terms “June” and “December” usage are better to avoid confusing.

L95, Ethical code is missing.

L124, incomplete sentence.

L125, 4-point Likert scale. What’s the scale anchors of 2 points and 3 points? The Likert scale is generally used by a 5-point design since the scale match the additivity and can be considered as a continuous variable, not just a category. The 4-point Likert scale is strange but may be appropriate to the study, however, the related references should be cited.

L129, Pain Catastrophizing Scale (PCS). The abbreviation PCS was mentioned in the following narratives.

L187, this subsection is fragmental. I suggest the contents can be merged up to one or two paragraphs.

L189, standard deviation (SD). The abbreviation SD was also mentioned in the following narratives.

L222, I suggest to present the data by mean (SD) years. The rest are inferred analogy.

L245, One point more on the Tampa Scale for Kinesiophobia would entail a 1.092 times higher chance of being on work-related leave. I do not think this is practically significant, even though statistically significant.

L247, In this manuscript, the tables were not good enough. I suggest that these should be redrawn to a better presentation.

L364, a paragraph regarding study limitation should be developed.

Reviewer 3 Report

Thank you very much for the opportunity to review the manuscript:  ‘Influence of psychosocial and sociodemographic variables on sickness leave and disability in patients with work-related spinal pain.’

In general, the topic of the manuscript appeared to be highly relevant to the field of study, with clear research questions.  Originality seemed assured for the most part but this aspect could have been emphasised further within the study’s description.  As far as I could ascertain, the study was based on sound conceptual and evidential theories.  For the most part, the methodology for this exploratory study was appropriate, although some additional details would have permitted the reader to appreciate the study’s limitations further.  The data analysis was thorough, although the rationale for some of the more ‘esoteric’ statistical analyses within an exploratory study, could have been made clearer.  The interpretation of the results was offered in a logical and informative way and the study’s discussion yielded some new perspectives on previously held assumptions and insights.  The manuscript was written well.

One or two aspects detracted from the manuscript’s overall quality and some of these specific points are detailed below for the authors’ consideration.

Abstract and Introduction.

line 9:  presumably the use of past tense might be helpful i.e.  . . . ‘is to describe’ should read ‘was to describe’?

line 10:   . . . spinal pain, to study sickness leave its duration, disability reported, . . . should read, ‘spinal pain, in order to study sickness leave and its duration, disability reported, . . .’

line 47: ‘medical’ should perhaps read ‘medical outcome’?

Methods.

line 77:  ‘is’ should read ‘was’.

line 81-83:  In regard to understanding the potential for bias or other such restrictions within the data, the authors might consider offering information as to whether n=129 was characterised by a sequential recruitment of all or some of the individuals matching inclusion criteria between the study’s dates?

line 94-96:  Although reference to ‘Helsinki’ and ‘organic law’ is mentioned by the authors, the reader might be interested in knowing whether or not specific institutional and/or regional health authority ethical opinion had been either sought or gained for this study.

line 191:  ‘after which’ should perhaps read ‘by which’?

line 199-200:  ‘the means of the clinical variables and age were compared using Student’s t test to see the differences between patients who presented sickness leave status and those who did not.’ perhaps should read ‘the means of the clinical variables and age were compared using Student’s t testing of the hypotheses of no differences between patients who presented sickness leave status and those who did not.’

Results.

line 201:  Did the authors mean to explore ‘association’ by using statistical modelling involving ‘regression’, with connotations of determination?

line 218:  ‘To increase comparative efficiency, the only subject under level 4 was grouped with those under level 3.’:  It could be argued that amalgamating categories in this way might increase the heterogeneity within ‘level 3’ and hinder this intended process?

line 230: It’s not immediately obvious from the data offered within Table 1 as to how when compared to ‘those with a university education, subjects with secondary education had a 4.44 times higher chance of going on sickness leave (CI 95%: 1.04-19.02). 231 The chance in subjects with primary education was 11.67 higher (CI 95%: 2.82-48.28).’?

line 250: Please amend Table 2’s heading to better reflect the description of data within the table i.e. Mean values for the different clinical variables as a function of the presence (yes) or absence (no) of sickness leave status.

line 262 (results) and earlier (line 187):  Within the authors’ exploratory modelling of correlates and determinants of sickness leave and disability, had checks been undertaken to ensure that the sample size and its dispersion characteristics on the variables of interest was sufficient to have offered robust statistical models?  Neither the conceptual model nor the experimental design sensitivity underpinning the approaches were explained in sufficient detail for the reader to fully appreciate the consequences of findings of ‘no association’ or indeed, spurious correlation given potentially, the relatively high number of independent variables.

line 271ff (Table 4) and line 286ff (Table 5):  Please use either ‘.’ or ‘,’ consistently as the separator of the whole and fractional number parts.

Discussion.

line 303ff   Do the sentiments expressed here ‘Consequently, the decision of whether to grant an individual leave of absence at a first consultation appears to be related to educational and occupational variables.’ act as a ‘non sequitur’ compared to what is offered immediately beforehand ‘. . .  no such association was observed between the duration of sickness absence and the educational and occupational level variables . . .’?  Please clarify/re-phrase.

line 306:  greater availability instead of ‘more ready availability’?

line 365:  Conclusions.  This section appears to offers predominantly a summary of findings rather than a contextualisation of their importance within the paper’s novel perspective of an occupational insurance provider and indeed, what future research might confirm or take forward.  It might be helpful to the reader for the reader to have been reminded of how the perspective of an occupational insurance provider, which is novel to this paper, differs from antecedent approaches. 
